# Effects of the Direction of Two Kirschner Wires on Combined Tibial Plateau Leveling Osteotomy and Tibial Tuberosity Transposition in Miniature Breed Dogs: An Ex Vivo Study

**DOI:** 10.3390/ani14152258

**Published:** 2024-08-03

**Authors:** Sanghyun Nam, Youngjin Jeon, Haebeom Lee, Jaemin Jeong

**Affiliations:** Department of Veterinary Surgery, College of Veterinary Medicine, Chungnam National University, Daejeon 34134, Republic of Korea; nmdanny86@gmail.com (S.N.); orangee0115@gmail.com (Y.J.); seatiger76@cnu.ac.kr (H.L.)

**Keywords:** tibia plateau leveling osteotomy, tibial tuberosity transposition, Kirschner wire angle, tension band technique, dog

## Abstract

**Simple Summary:**

This study investigates how the direction of wire placement affects the stability of combined tibial plateau leveling osteotomy and tibial tuberosity transposition surgery in miniature-breed dogs suffering from two common stifle problems: a torn cruciate ligament and patella luxation. Using the bones from 21 small dogs, we tested pin placement in two directions: proximal (proximal osteotomized segment) and distal. This study also tested different wire thicknesses and the use of tension bands. The results showed that distal pin placement provided better stability in the 0.56 mm tension band group. This finding is important because it offers a solution for cases where there is not enough space to place pins in the proximal direction. In situations where there is inadequate space for proximal pin placement, distal pin placement can achieve similar stability. This research provides valuable insights that can lead to better surgical outcomes for small animal surgeons.

**Abstract:**

This study evaluates the impact of Kirschner wire (K-wire) insertion direction on the biomechanical properties of combined tibial plateau leveling osteotomy (TPLO) and tibial tuberosity transposition (TTT) procedures in small-breed dogs with cranial cruciate ligament rupture and medial patella luxation. Twenty-one cadaveric tibiae were divided into two groups; the specimens were divided into two groups; one underwent TPLO-TTT with a proximal pin placement (Group TTP), and the other received TPLO-TTT with a distal pin placement (Group TTD). For both pin placements, two additional subgroups were formed: one with a 0.56 mm tension band (Groups TTP0.56 and TTD0.56) and the other with a 0.76 mm tension band (Groups TTP0.76 and TTD0.76). The tensile force was applied, and failure load and mode were recorded. The distal pin direction in Group TTD0.56 exhibited a significantly higher mean failure load (380.1 N) compared to the proximal pin direction in Group TTP0.56 (302.2 N, *p* = 0.028). No significant differences were observed among the other groups. This study concludes that distal pin placement can provide similar or improved mechanical stability in cases with limited space for proximal pin placement during combined TPLO and TTT procedures.

## 1. Introduction

Medial patellar luxation (MPL) and cranial cruciate ligament rupture (CCLR) represent predominant causes of hindlimb lameness within the canine population, exhibiting a particularly high prevalence among small-breed dogs [1,2,3,4]. According to previous studies, small breeds are significantly more susceptible to MPL, with a reported incidence rate up to 12 times higher than that observed in larger breeds [5,6]. Moreover, the occurrence of concurrent CCLR in patients diagnosed with MPL ranges from 22% to 44%, underscoring the clinical complexity and necessitating effective surgical interventions to restore limb function [7,8].

Surgical techniques for treating MPL commonly include a variety of techniques such as soft-tissue realignment, trochleoplasty, tibial tuberosity transposition (TTT), and distal femoral osteotomy [4]. The TTT method, specifically, addresses congenital or acquired malalignments of the quadriceps mechanism, a condition frequently accompanying MPL. A TTT is a common procedure performed in surgical correction of the medial patellar luxation in dogs [2,4]. This technique involves the osteotomy and subsequent repositioning of a tibia tuberosity segment to counteract the luxation direction, secured in its new location with surgical implants [1,3,4,9]. In cases of CCLR, procedures like tibia plateau leveling osteotomy (TPLO), CORA-based leveling osteotomy (CBLO), and tibia tuberosity advancement (TTA) are integrated to effectively address joint instability [10,11].

Recent studies have reported several methods that address concurrent MPL and CCLR within a single session. These methods, including modified TPLO (mTPLO), tibia tuberosity transposition advancement (TTTA), and combined TPLO with TTT, aim to address quadriceps malalignment and stabilize the CCL-deficient stifle joints [10,11,12,13,14]. Various techniques have been explored to improve the mechanical stability and clinical outcomes of these procedures. Although previous ex vivo studies evaluating combined TPLO-TTT report that TPLO-TTT has weaker mechanical stability compared to TTT or TPLO alone [15], it has been shown to provide relatively good short- to long-term outcomes in retrospective clinical studies [11,16,17].

However, performing combined TPLO-TTT in miniature-breed dogs can be challenging in surgical precision, particularly with the insertion of Kirschner wire (K-wire) for TTT fixation. This difficulty mainly arises from the impingement of pins on TPLO screws in the rotated proximal tibial plateau segment. Alternatively, K-wires can be inserted in the distal direction instead of the proximal direction into the proximal tibial plateau segment. However, there is a lack of studies on how the biomechanical properties of TTT fixation in TPLO-TTT change when the insertion direction of these K-wires is altered.

Therefore, the purpose of this study is to analyze the biomechanical effects of the Kirschner wire insertion angle (KWIA), in the combined TPLO-TTT technique in 2–5 kg miniature-breed dogs. Our hypothesis was that there would be no difference in load at failure or mode of failure between the direction of the K-wire, regardless of whether a tension band (TB) was applied or the size of the TB wire if applied.

## 2. Materials and Methods

### 2.1. Specimens and Preparation

Twenty-one donated miniature-breed cadavers weighing between 2 and 5 kg, euthanatized for reasons unrelated to this study, were included in the ex vivo study. The ethics approval for the cadaveric study protocol was not required by the Institutional Animal Care and Use Committee of Chungnam National University. Cadavers were preserved at −20 °C and thawed to room temperature (21 °C) prior to experimentation. Comprehensive craniocaudal and mediolateral radiographic of each tibia were performed with a 25 mm radiographic marker for precise measurement. The specimens were then divided into six groups based on the specific surgical technique applied and the size of the tension band (TB). Each group included 6 tibiae, representing either the left or right hindlimb from one individual dog.

The groups were divided as follows: Group TTP: TPLO + TTT with the proximal insertion of two K-wires, no tension band; Group TTD: TPLO + TTT with the distal insertion of two K-wires, no tension band; Group TTP0.56: TPLO + TTT with the proximal insertion of two K-wires and a 0.56 mm tension band; Group TTD0.56: TPLO + TTT with the distal insertion of two K-wires and a 0.56 mm tension band; Group TTP0.76: TPLO + TTT with the proximal insertion of two K-wires and a 0.76 mm tension band; and Group TTD0.76: TPLO + TTT with the distal insertion of two K-wires and a 0.76 mm tension band.

A coin flip method was used to assign the left or right hindlimb of each group. For example, if the left hindlimb was assigned to the TTP group, the right hindlimb from the same dog was assigned to the TTD group, ensuring that each dog provided paired samples for comparison. The two K-wires were inserted parallel to each other at the same level. This process was similarly applied to the TTP0.56 group (TTP + 0.56 mm TB) and the TTD0.56 group (TTD + 0.56 mm tension band wire), as well as the TTP0.76 group (TTP + 0.76 mm tension band wire) and the TTD0.76 group (TTD + 0.76 mm tension band wire) (Figure 1). The remaining three pairs of the hindlimb were included as a control group, which only underwent TPLO.

### 2.2. Surgical Techniques

Preoperative TPLO planning was performed using digital software vPOP (vPOP Pro version 2.9.6, VetSOS Education Ltd., Shrewsbury, UK). TPLO and TTT were conducted on all 36 tibiae following established protocols from a previous study [9,18]. A 12 mm crescentic saw blade was used for the osteotomy, placing the intercondylar eminence as the central point. Temporary fixation was performed with a 1.0 K-wire after rotation of the tibial plateau segment and achieving a tibial plateau angle (TPA) of 6°, and a 2.0 mm TPLO bone plate (Able Inc., Jeonju, Republic of Korea) was used for osteotomy stabilization [19]. Following plate application, TTT was performed using an oscillating saw (DePuy Synthes Vet, Oberdorf, Switzerland) to ensure a linear osteotomy and to preserve the craniodistal periosteal attachment. The osteotomized segment was laterally translated into a distance of one-third of the tibia tuberosity’s width. K-wires were inserted at the proximal insertion point of the patellar ligament to secure the segment, passing through and emerging from the opposite tibial cortex. TB was applied as a single twist to the TTP-0.56, TTD-0.56, TTP-0.76, and TTD-0.76 groups.

### 2.3. Radiographic Measurements

Orthogonal radiographs of each pair of tibiae were obtained to assess the pin insertion angle and postoperative TPA using commercially available radiographic software (Zetta PACS Viewer 2001, Taeyoung Soft Co., Ltd., Gwacheon, Republic of Korea). K-wire insertion angles were determined on lateral radiographs by measuring the angle between the osteotomy line and the inserted K-wire, as described in a previous study (Figure 2) [20]. The two K-wires were assigned as P1 and P2, and the data were obtained as the mean standard deviation of the K-wire angle by calculating [P1 + P2]/2.

### 2.4. Mechanical Testing

Specimens were mounted in a 35 mm internal diameter stainless steel pipe using methyl methacrylate resin (Trayplast, Vertex, Soesterberg, The Netherlands) and then attached to a loading cell within a custom-designed jig. The tibiae were then secured to the load cell using a custom-designed jig. The patellar tendon was secured to the actuator using a custom-made clamp (Figure 3). The jig was configured to hold the tibia and patellar ligament at 135 degrees to mimic the midstance angle of the stifle of dogs [21]. A testing machine (WL2100C, WITHLAB Co., Ltd., Gunpo, Republic of Korea) was used to perform vertical distraction at a displacement rate of 20 mm/min until failure of the tibia construct was observed. The load at failure (Newtons) and the mode of failure were recorded for each specimen. All testing was video-recorded (iPhone 14, Apple Inc., Cupertino, CA, USA) with the load display and tibias on the same screen (Figure 3).

### 2.5. Statistical Analysis

Based on the results of a previous study [20], a prospective power analysis was conducted using statistical software (G*Power V3.1.9.2x Dusseldorf, Germany), and it was determined that a sample size of 6 constructs (*n* = 6 tibia/group) would have alpha = 0.05, power = 0.8 and an estimated effect size (ES, d = 1.4790362). Statistical analysis was performed using commercially available software (IBM SPSS Statistics 24.0, IBM Corp., Chicago, IL, USA). Data were nonparametrically distributed, and postoperative K-wire angle and load at failure were evaluated between the groups using the Wilcoxon test and Kruskal–Wallis test. The Mann–Whitney U test was used for post hoc analysis. Statistical significance was set at a value of *p* < 0.05.

## 3. Results

All cadavers met the criteria with a total of 36 stifles and were skeletally mature. The median body weight was 3.6 kg (range: 2–5 kg), and there were 13 intact females, 2 spayed females, and 3 intact males. The most dominant breed was Poodle (*n* = 9), followed by Maltese (*n* = 4), Bichon (*n* = 2), Chihuahua (*n* = 1), and Pomeranian (*n* = 1). Descriptive data, including body weight, failure modes, preoperative tibial plateau angle, postoperative plateau angle, and the K-wire insertion number, are described in Table 1. Biomechanical test results and KWIA are described in Table 2.

In the comparison of the proximal pin load at failure, TTP0.56 exhibited significantly greater strength compared to TTP (*p* = 0.016). Similarly, TTP0.76 demonstrated significantly greater strength than TTP (*p* = 0.037). However, the difference between TTP0.56 and TTP0.76 was not statistically significant, although it was close to significance (*p* = 0.078). For the distal pin load at failure, TTD0.56 showed significantly greater strength compared to TTD (*p* = 0.004). Additionally, TTD0.56 demonstrated significantly greater strength than TTD0.76 (*p* = 0.01). However, the difference between TTD and TTD0.76 was not significant (*p* = 0.109).

In the TTP and TTD groups, in which a TB was not applied, tibial tuberosity avulsion failure was observed (Figure 4). However, fractures were observed in some of the 0.56 tension band (TB) groups (TTP0.56 and TTD0.56) and all the 0.76 tension band (TB) groups (TTP0.76 and TTD0.76) as failure modes. The fracture occurred at the level where the tibia was embedded in the resin, precisely at the most superficial interface between the two materials. The number of fractures observed per group is specified in Table 1. In specimens that had more than one trial of proximal K-wire insertion, pin bending (Figure 5A) and locking screw damage was observed (Figure 5B). In the proximal pin groups (TTP, TTP0.56, and TTP0.76), damaged locking screws were found in proximal locking screws, whereas in the distal pin groups (TTD, TTD0.56, and TTD0.76), no screw damage was observed.

The comparison of KWIA between the proximal pin groups and the distal pin groups was significant (*p* = 0.000). The TTP and TTD groups showed a significant difference (*p* = 0.028), as did the TTP0.56 and TTD0.56 groups (*p* = 0.028); similarly, the TTP0.76 and TTD0.76 groups showed a significant difference (*p* = 0.028).

## 4. Discussion

This study evaluated the biomechanical properties of combined TTT and TPLO depending on the direction of the Kirschner wire insertion angle and the application of the tension band. Our hypothesis posited that there would be no significant difference in the biomechanical integrity, load at failure, or mode of failure between proximal pin placement and distal pin placement. This hypothesis was partially accepted, as groups TTP, TTD, TTP0.76, and TTD0.76 showed no difference in load at failure and mode of failure. However, between the TTP0.56 and TTD0.56 groups, a significant difference was observed in the load at failure, with the distal pin group showing higher mechanical strength.

According to previous studies [20,22,23,24], inserting the pin in a caudodistal or transverse direction is known to provide better counteraction against the patellar tendon force during TTT surgery. Additionally, the AO principle for TB to manage fractures recommends the placement of two parallel pins across and perpendicular to the fracture plane [25]. However, in the current study, inserting the K-wire perpendicular to the TTT osteotomy line and into the proximal tibial plateau segment to stabilize the osteotomized tuberosity segment posed a challenge. This difficulty arose because of the conflict between the pins and the proximal locking screw due to the narrow space in the proximal tibial plateau segment, especially in miniature breed dogs. Therefore, in the proximal pin groups, the pin was inserted at a mean angle of 74.23 degrees to the osteotomy line.

Inserting the K-wire into the proximal tibial plateau segment caused resistance with the proximal locking screw, resulting in multiple insertion attempts in groups TTP, TTP0.56, and TTP0.76. Although the wire eventually flexed slightly and passed through, this process caused damage to the threads of the locking screw (Figure 5A). Additionally, these attempts increased the space where the K-wire passed through, which can lead to implant loosening in a clinical setting.

The failure mode observed in groups TTP, TTD, TTP0.56, and TTD0.56 was similar to that reported in studies on TPLO-TTT with or without a tension band, involving distal tibial crest displacement and rupture of the patellar ligament [15,26]. However, fracture occurred at the diaphysis of the tibia in four specimens, with three in TTP0.56 and one in TTD0.56 groups. This failure mode was consistent with observations from a previous TTT study [23]. The occurrence of such failures may be attributed to histological damage during freeze–thaw process of the specimens or differences in bone quality between specimens. However, if such failures were to occur in a clinical setting, they could have catastrophic consequences. Therefore, further research is necessary to assess the stress distribution in TPLO-TTT constructs with and without a tension band.

One of the interesting findings in the current study was that the failure mode in the TTP0.76 and TTD0.76 groups involved fractures at the junction between the tibia and the resin. In contrast to the 0.56TB group, where two out of the four fractured specimens experienced distal periosteum breakage, the specimens in the 0.76TB group all had intact distal periosteum. Although the actual mechanical test results showed that the 0.56TB specimens performed better than the 0.76TB specimens, the test results and failure modes may be attributed to the stress shielding effect [27,28] or stress concentration occurring at the junction where the tibia and cement, which have different material properties [29,30,31]. This result presents a clinical dilemma when choosing the appropriate tension band size in practice. Despite the expectation that the thicker 0.76TB would withstand higher mechanical loads, tibia diaphyseal fractures occurred at lower loads compared to the avulsion fractures observed in the 0.56TB group. These findings are in contradiction with the results reported by Neat and colleagues, which showed that larger TB sizes generally provide higher mechanical strength in an olecranon osteotomy model [32]. Therefore, further research is needed to determine the appropriate TB size based on bone size or patient weight. Additionally, it should be noted that this study used only one twist knot for securing the tension band wire due to the small bone size. According to a previous study, placing two knots results in more rigid fixation to the construct compared to a single knot [33]. One twist knot may not provide uniform tension across the entire wire, especially in larger bones, potentially resulting in less effective tightening compared to using two twist knots.

According to a previous study, the approximate quadriceps muscle force during walking for dogs in the weight range of the current study is 33.68 N [34]. All groups in our experiment, regardless of TB application and pin direction, exhibited forces higher than this value. However, in the TTP and TTD groups in which only pins were applied, the load at failure ranged from 39.37 N to 301.11 N, suggesting a potential risk of construct failure under conditions of higher forces such as those encountered during trotting. In contrast, the 0.56 TB groups showed a load at failure ranging from 227.56 N to 480.03 N, and the 0.76 TB groups exhibited a load at failure ranging from 164.16 N to 349.46 N. These results suggest significantly improved mechanical stability when a TB is applied. Therefore, it is cautiously recommended to apply TBs during TPLO-TTT in a clinical setting.

Our study has limitations, and caution is required for direct clinical application. First, since our study was designed ex vivo, with muscles and ligaments removed from the tibia except for the quadriceps muscle and patellar tendon. Therefore, the results may not fully replicate the actual weight-bearing situation. In clinical settings, the soft tissue and callus formation around the proximal tibia may provide additional resistance. Secondly, since all surgical procedures were performed by a single surgeon, outcomes related to experience may vary, including a potential bias where the 0.56 mm wire might have been tightened more stably due to ease of manipulation compared to the thicker 0.76 mm wire when applying the TB. Thirdly, we could not clarify the mechanism of fracture observed in the 0.76 TB groups. Although the results showed no significant differences based on pin direction in the 0.76 TB groups, further study is necessary, as different outcomes may arise when conducting the procedure on larger bone specimens.

## 5. Conclusions

In conclusion, our study primarily aimed to analyze the biomechanical effects of the Kirschner wire insertion angle in the combined TPLO-TTT technique for 2–5 kg miniature-breed dogs. Our finding suggests that, in general, the direction of the K-wire does not significantly affect stability, except when using a 0.56 mm tension band. When a 2.0 mm TPLO plate and a 0.56 TB are used for the combined TPLO and TTT, inserting the K-wire distally may be considered if proximal pin insertion is challenging due to the proximal locking screw. Furthermore, significant differences were observed when a TB was not applied during TPLO-TTT compared to the TB-applied group. Therefore, in clinical settings, the application of TBs is recommended during TPLO-TTT to avoid the risk of construct failure.

## Figures and Tables

**Figure 1 animals-14-02258-f001:**
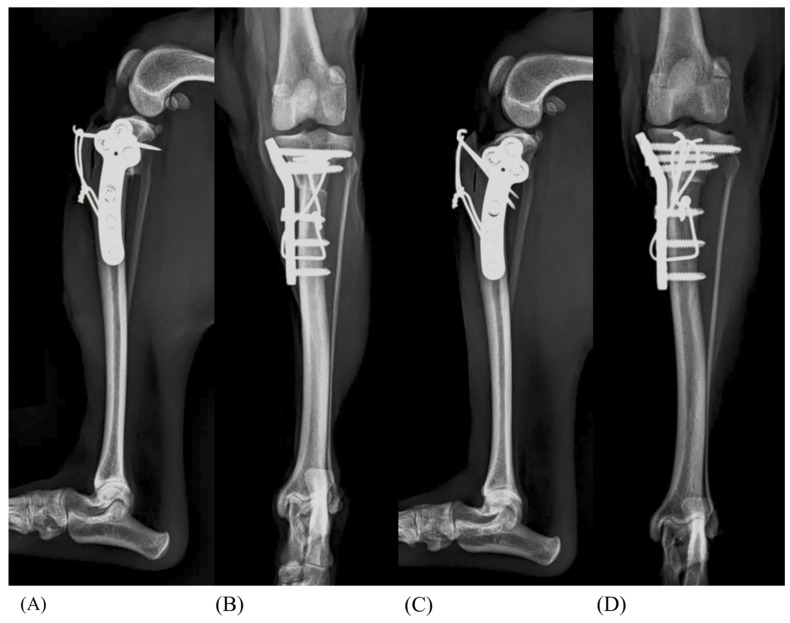
Craniocaudal and mediolateral radiographs of the TTP0.76 (**A**,**B**) and TTD0.76 (**C**,**D**) groups. The tension band was applied in a figure-eight pattern, and the pins were 1.0 mm and 0.8 mm.

**Figure 2 animals-14-02258-f002:**
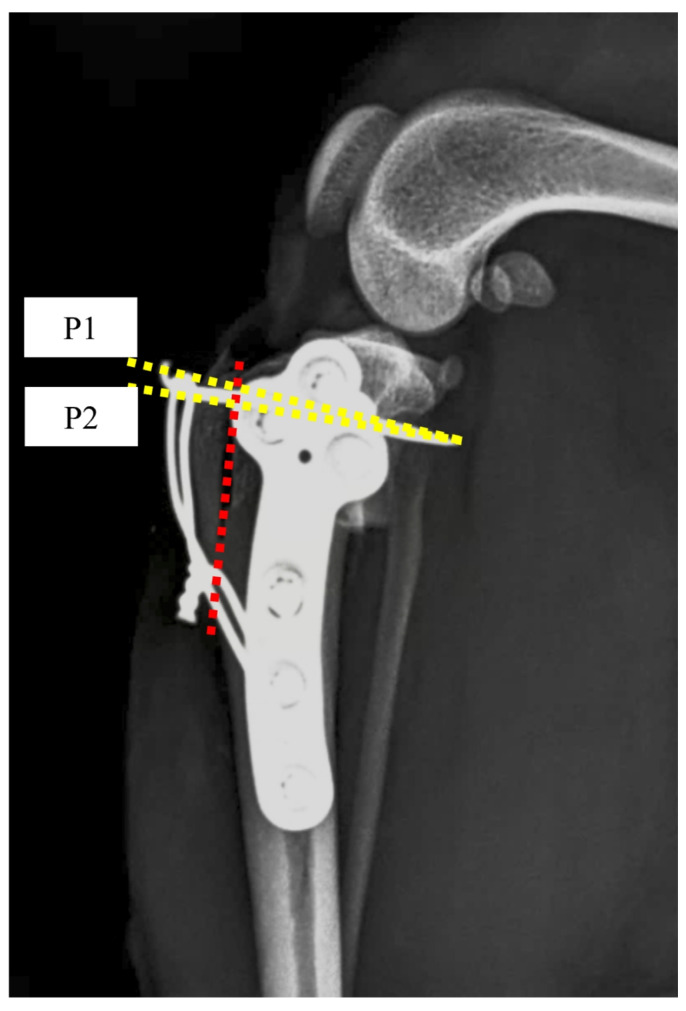
K-wire insertion angle (KWIA) was measured using Zetta PACS (Taeyoung Soft Co., Ltd., Gwacheon, Republic of Korea). The angle formed by the K-wire and the osteotomy line (red dotted line) was measured. The two K-wires were assigned as P1 and P2 (yellow dotted line), and the data were obtained as the mean standard deviation of the K-wire angle by calculating [P1 + P2]/2.

**Figure 3 animals-14-02258-f003:**
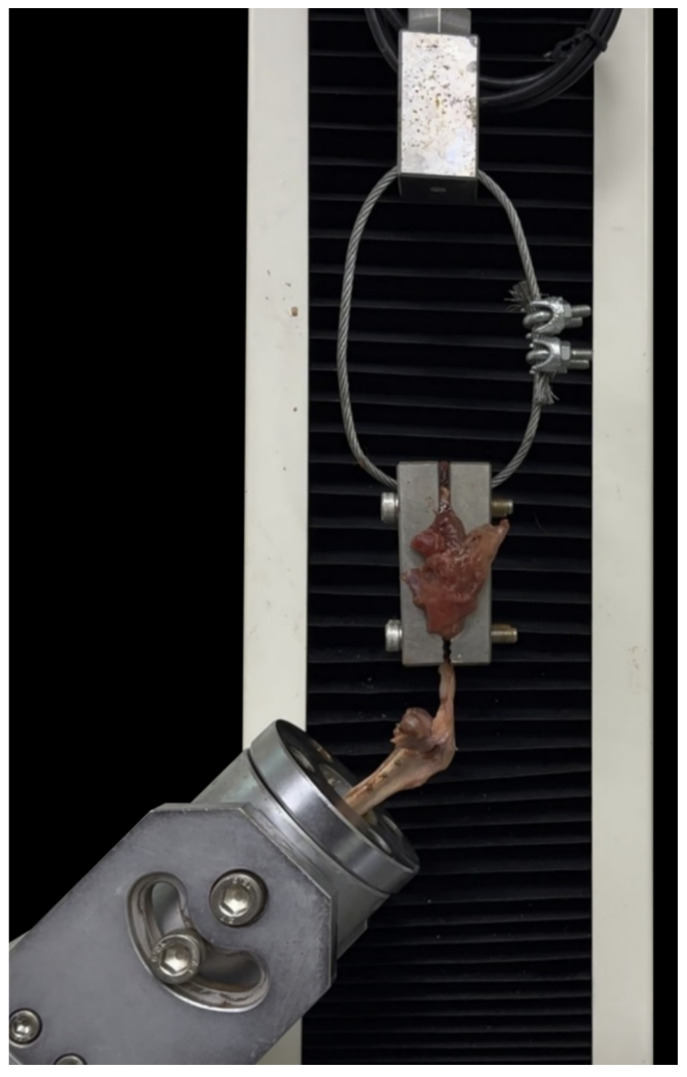
Biomechanical testing setup. Specimens were mounted using methyl methacrylate resin (Trayplast, Vertex, Soesterberg, The Netherlands) in a stainless steel pipe with an inner diameter of 35 mm. The specimens were secured to the loading cell, and the patellar tendon was fixed at an angle of 135° in a custom-designed clamp.

**Figure 4 animals-14-02258-f004:**
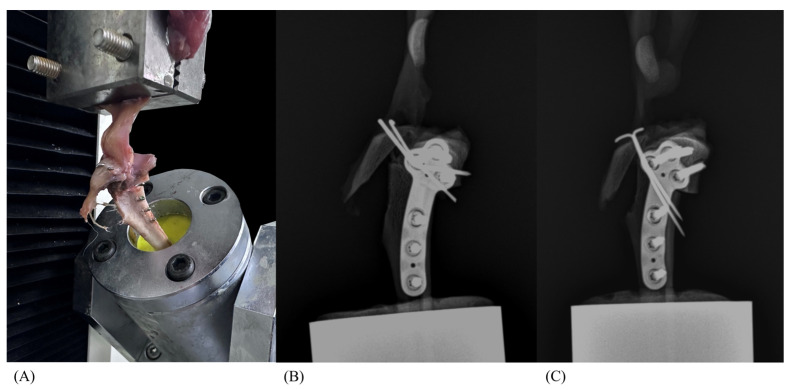
Photograph of tibial tuberosity avulsion failure mode (**A**) and mediolateral radiograph of TTP (**B**) and TTD (**C**). Distal periosteum breakage is observed, along with bending of both the proximal pin and distal K-wires.

**Figure 5 animals-14-02258-f005:**
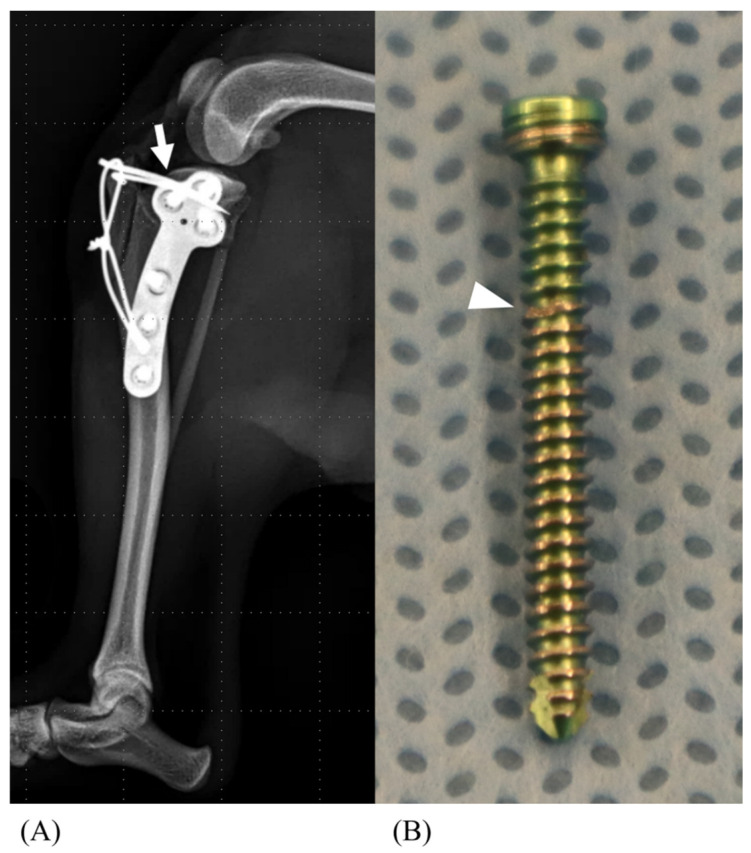
Mediolateral radiograph of the proximal pin group (**A**). During pin insertion, the pin bends due to the limited space caused by the proximal tibial plateau segment locking screw (arrow). The photograph of the damaged proximal locking screw due to K-wire insertion (**B**).

**Table 1 animals-14-02258-t001:** Comparison of body weight, tibial plateau angle (TPA), pin insertion number, and failure mode.

	Body Weight (Kg)	Preoperative TPA	Postoperative TPA	Pin Insertion Number	Failure Mode
Group 1 TTP	3.62 ± 0.38	28.90 ± 4.35°	6.33 ± 2.34°	0 (*n* = 1),1 (*n* = 2), 2 (*n* = 2), 3 (*n* = 1)	Avulsion (*n* = 6)
Group 2 TTD	3.62 ± 0.38	28.67 ± 3.37°	5.33 ± 1.51°	0 (*n* = 5), 1 (*n* = 1)	Avulsion (*n* = 6)
Group 3 TTP0.56	3. 92 ± 0.71	30.70 ± 3.16°	6.33 ± 2.73°	0 (*n* = 1), 1 (*n* = 2), 2 (*n* = 3)	Avulsion (*n* = 3), Fracture (*n* = 3)
Group 4 TTD0.56	3. 92 ± 0.71	30.13 ± 3.07°	5.83 ± 2.99°	0 (*n* = 5), 1 (*n* = 1)	Tendon rupture (*n* = 2), Avulsion (*n* = 3), Fracture (*n* = 1)
Group 5 TTP0.76	3.34 ± 0.37	30.63 ± 3.13°	6.67 ± 2.07°	1 (*n* = 2), 2 (*n* = 1), 3 (*n* = 2)	Fracture (*n* = 6)
Group 6 TTD0.76	3.34 ± 0.37	30.20 ± 3.12°	6.33 ± 1.51°	0 (*n* = 6)	Fracture (*n* = 6)
Control TPLO	3.45 ± 0.45	25.47 ± 6.45°	7.17 ± 1.33°		Fracture (*n* = 6)

TTP, TPLO-TTT with proximal pin; TTD, TPLO-TTT with distal pin; TTP0.56, TPLO-TTT with proximal pin and 0.56 tension band; TTD0.56, TPLO-TTT with distal pin and 0.56 tension band; TTP0.76, TPLO-TTT with proximal pin and 0.76 tension band; TTD0.76, TPLO-TTT with distal pin and 0.76 tension band.

**Table 2 animals-14-02258-t002:** Comparison of load at failure, yield force, periosteal bridge failure, and K-wire insertion angle by group.

	Group 1 TTP	Group 2 TTD	Group 3 TTP0.56	Group 4 TTD0.56	Group 5 TTP0.76	Group 6 TTD0.76	TPLO Control
Load at failure (N)	124.23 ±102.62	142.54 ±97.01	302.24 ±52.92 ^a^	380.15 ±58.30 ^a^	255.20 ±44.50	241.07 ±66.41	242.38 ±31.98
Yield force (N)	119.36 ±107.05	132.61 ±105.61	229.48 ±126.57	282.53 ±100.06	210.42 ±32.22	172.24 ±32.55	214.23 ±69.95
Periosteal bridge failure	6/6	6/6	3/6	5/6	N/A	N/A	N/A
K-wire insertion angle (°)	72.58 ±3.71 ^b^	48.52 ±9.14 ^b^	73.81 ±3.87 ^a^	46.94 ±11.87 ^a^	76.26 ±8.16 ^c^	47.48 ±2.54 ^c^	N/A

TTP, TPLO-TTT with proximal pin; TTD, TPLO-TTT with distal pin; TTP0.56, TPLO-TTT with proximal pin and 0.56 tension band; TTD0.56, TPLO-TTT with distal pin and 0.56 tension band; TTP0.76, TPLO-TTT with proximal pin and 0.76 tension band; TTD0.76, TPLO-TTT with distal pin and 0.76 tension band; ^a^ *p*-value between Group 3 and 4 showed a significant difference, ^b^ *p*-value between Group 1 and 2 showed a significant difference, ^c^ *p*-value between Group 5 and 6 showed a significant difference; *p* < 0.05 was set as significant difference.

## Data Availability

The original data used in this study are included in the article; further inquiries can be directed to the corresponding author.

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
