# Peer review of "Effects of the Direction of Two Kirschner Wires on Combined Tibial Plateau Leveling Osteotomy and Tibial Tuberosity Transposition in Miniature Breed Dogs: An Ex Vivo Study"

_animals, 2024, doi:10.3390/ani14152258_

Round 1

Reviewer 1 Report

Comments and Suggestions for Authors

Effects of Direction of Two Kirschner Wires on Combined Tib- 2 ial Plateau Leveling Osteotomy and Tibial Tuberosity Transpo- 3 sition in Miniature Breed Dogs: an Ex vivo Study

  • A brief summary 

This paper explores the potential mechanical implications of K-wire direction in concurrent TPLO and TTT surgeries in small breed dogs suffering from cruciate disease and medial patellar luxation. The study was designed as an ex-vivo cadaveric study with 7 groups. The outcomes were determined based on peak failure force of the constructs.

  • General concept comments
    Article:

The general concept of the paper is sound. There were no methodological inaccuracies observed.

  • The manuscript is clear, relevant to the field and presented in a well-structured manner.
  • The reference (13) is a self-citation, but the cited article is an original research article regarding a similar problem challenging the same clinical problem, with a different technique.
  • The manuscript is scientifically sound and is the experimental design is appropriate to test the hypothesis.

The manuscript’s results are reproducible based on the details described in the methods section.

  • Figures and tables are well presented and explained.
  • The conclusions are consistent with the evidence and arguments presented.

Author Response

Comments 1: Effects of Direction of Two Kirschner Wires on Combined Tibial Plateau Leveling Osteotomy and Tibial Tuberosity Transposition in Miniature Breed Dogs: an Ex vivo Study

A brief summary

This paper explores the potential mechanical implications of K-wire direction in concurrent TPLO and TTT surgeries in small breed dogs suffering from cruciate disease and medial patellar luxation. The study was designed as an ex-vivo cadaveric study with 7 groups. The outcomes were determined based on peak failure force of the constructs.

General concept comments

Article:

The general concept of the paper is sound. There were no methodological inaccuracies observed.

The manuscript is clear, relevant to the field and presented in a well-structured manner.

The reference (13) is a self-citation, but the cited article is an original research article regarding a similar problem challenging the same clinical problem, with a different technique.

The manuscript is scientifically sound and is the experimental design is appropriate to test the hypothesis.

The manuscript’s results are reproducible based on the details described in the methods section.

Figures and tables are well presented and explained.

The conclusions are consistent with the evidence and arguments presented.

Response 1: Thank you for your positive feedback on our manuscript. Your recognition is greatly appreciated.

Reviewer 2 Report

Comments and Suggestions for Authors Authors evaluated the effect of the direction of TB pins, proximal or distal on stability of TPLO+TTT. It is well known principle pin inserts perpendicular to the tension line in tension band wiring. To be scientific it is needed to group with angles to the tension line.

Reviewer 3 Report

Comments and Suggestions for Authors

Thank you for the paper. The study is well designed and conducted.

I have some minimal comment for you that you can find in the enclosed document.

Line 88: Can you precise that pins are inserted at the same level (because some surgeons place pins “vertically aligned”)

Line 113: can you just precise that you have only one twist knot

Line 124: just add “on lateral radiographs” after “determined”

Line 125: it is figure 2 not 1.

Line 133: it is figure 3 not 2.

Line 158: intact male or castrated?

Line 187: I don’t understand this sentence. Fracture occurred at the level of attachment of the tibia with steel pipe, is that correct? If it is, can you be more specific.

Line 239: Can you modify “cement” by “resin” because you use the term resin previously.

Line 249: can you add a comment about the fact that you have only one twist knot. Because tightening only one may not tighten the whole wire (or less tighten than 2).

Can you also add a comment that maybe it is easier to tight the small one.

I will add probably a reference to say that your results are in contradiction with this paper: Neat BC, Kowaleski MP, Litsky AS, Boudrieau RJ. Mechanical evaluation of pin and tension-band wire factors in an olecranon osteotomy model. Vet Surg. 2006 Jun;35(4):398-405.

Reviewer 4 Report

Comments and Suggestions for Authors

I appreciate the opportunity to revise the manuscript on evaluating the effects of the direction of Kirschner Wires on Combined TPLO and TTT in miniature breed dogs. The results indicated no significant differences in the stability with either direction of the K-wire insertion.

Based on the revision, here are specific suggestions:

1) The study design could be more specifically and appropriately presented. Greater detail and accuracy are necessary to understand exactly how many specimens and what exact intervention took place in each group.

2) Define abbreviations like KWIA before their first use in the manuscript.

3) In the results section Lines 157-158 please specify the exact number of male dogs.

4) Figure 5b: a more indicative photograph of the damaged proximal screw would be appreciated.

5) Conclusions have focused more on the effects of tension band  during TPLO-TTT (3.5 lines) that it is not reported to be a questiion of the study in the Introduction section as authors state that the purpose of that study wast o analyze the biomechanical effects of Kirschner wire insertion angle in the combined TPLO-TTT technique in 2-5kg toy-breed dogs. Their hypothesis was that there would be no difference in load at failure or mode of failure between the direction of the K-wire and for that question the conclusion extent is 4.5 lines.

Reviewer 5 Report

Comments and Suggestions for Authors

Dear Authors,

The article is easy to understand. However, more information about the article can be given in the introduction section. There are many studies especially on TPLO and TTT techniques. These articles can also strengthen the discussion section.

Regards

Round 2

Reviewer 4 Report

Comments and Suggestions for Authors

Ι would like to thank the authors for their satisfactory editing of their manuscript.  This version is improved. Congratulations!